# A Single Bout of Ultra-Endurance Exercise Reveals Early Signs of Muscle Aging in Master Athletes

**DOI:** 10.3390/ijms23073713

**Published:** 2022-03-28

**Authors:** Cécile Coudy-Gandilhon, Marine Gueugneau, Christophe Chambon, Daniel Taillandier, Lydie Combaret, Cécile Polge, Guillaume Y. Millet, Léonard Féasson, Daniel Béchet

**Affiliations:** 1Université Clermont Auvergne, INRAE, UNH, Unité de Nutrition Humaine, CRNH Auvergne, F-63000 Clermont-Ferrand, France; cecile.coudy-gandilhon@inrae.fr (C.C.-G.); marine.gueugneau@inrae.fr (M.G.); daniel.taillandier@inrae.fr (D.T.); lydie.combaret@inrae.fr (L.C.); cecile.polge@inrae.fr (C.P.); 2Université Clermont Auvergne, INRAE, Metabolomic and Proteomic Exploration Facility, F-63000 Clermont-Ferrand, France; christophe.chambon@inrae.fr; 3Université de Lyon, UJM-Saint-Etienne, Inter-University Laboratory of Human Movement Biology, EA 7424, F-42023 Saint-Etienne, France; guillaume.millet@univ-st-etienne.fr (G.Y.M.); leonard.feasson@univ-st-etienne.fr (L.F.)

**Keywords:** aging, exercise, skeletal muscle, capillaries, lipid droplets, extracellular matrix

## Abstract

Middle-aged and master endurance athletes exhibit similar physical performance and long-term muscle adaptation to aerobic exercise. Nevertheless, we hypothesized that the short-term plasticity of the skeletal muscle might be distinctly altered for master athletes when they are challenged by a single bout of prolonged moderate-intensity exercise. Six middle-aged (37Y) and five older (50Y) master highly-trained athletes performed a 24-h treadmill run (24TR). *Vastus lateralis* muscle biopsies were collected before and after the run and assessed for proteomics, fiber morphometry, intramyocellular lipid droplets (LD), mitochondrial oxidative activity, extracellular matrix (ECM), and micro-vascularisation. Before 24TR, muscle fiber type morphometry, intramyocellular LD, oxidative activity, ECM and micro-vascularisation were similar between master and middle-aged runners. For 37Y runners, 24TR was associated with ECM thickening, increased capillary-to-fiber interface, and an 89% depletion of LD in type-I fibers. In contrast, for 50Y runners, 24TR did not alter ECM and capillarization and poorly depleted LDs. Moreover, an impaired succinate dehydrogenase activity and functional class scoring of proteomes suggested reduced oxidative phosphorylation post-24TR exclusively in 50Y muscle. Collectively, our data support that middle-aged and master endurance athletes exhibit distinct transient plasticity in response to a single bout of ultra-endurance exercise, which may constitute early signs of muscle aging for master athletes.

## 1. Introduction

Skeletal muscle is central for not only coordinated movements and postural control but also for energy metabolism [1] and myokines secretion [2]. Skeletal muscle is the most abundant tissue in the adult body and a major storage site for amino acids (in the form of myofibrillar proteins) and glucose (in the form of glycogen). Skeletal muscle thereby plays an important role in many physiological processes, including carbohydrate metabolism, fatty acid oxidation and thermogenesis [3].

A striking physiological characteristic of the skeletal muscle is also its capacity to progressively modulate local blood flow, substrate utilization, energy production and contractile proteins in response to long-term endurance training [4]. Such adaptative changes are important to blunt the homeostatic threats generated by exercise challenges and to promote optimal performance. The predominant fuels used during endurance exercise are fats and carbohydrates. Carbohydrates are stored as muscle and liver glycogen, and fats are stored as reserves in subcutaneous and visceral adipose tissue. Smaller quantities of fats are present in circulating lipoprotein particles and in lipid droplets inside muscle fibers. For endurance athletes, long-term adaptations of the skeletal muscle to aerobic training are reflected by gradual enhancements in capillary density and transport capacity of fatty acids and glucose, increases in metabolic enzyme activities and mitochondrial density, an accumulation of intramyocellular lipid droplets (LD), and adaptations of contractile proteins to slow type isoforms [5]. Changes in functional and morphological characteristics of the skeletal muscle also involve a remodeling of the extracellular matrix (ECM) embedding muscle fibers [6].

Such adaptive changes of the skeletal muscle to long-term endurance training nonetheless enable short-term plasticity. Even for highly-trained endurance athletes, a single bout of exercise still elicits transient changes in myocellular processes, reflecting metabolic and functional adaptation capacities. Accordingly, current evidence demonstrates short-term malleability of intramyocellular lipids [7], and remodeling of muscle mitochondrial and proteolytic pathways [8] in response to a single bout of aerobic exercise. Whether such short-term plasticity also occurs for the ECM and capillarization remains to be established.

The loss of skeletal mass and function during the aging process (sarcopenia) is one of the most dramatic changes affecting the human body. Most previous studies investigating human sarcopenia relied on the comparison between young (e.g., 20–30 years) and old subjects (65–75 years). They provided major information about changes in fiber morphology, oxidative metabolism, lipid droplets [9], capillarization, ECM fibrosis [10], ions and oxylipins homeostasis [11,12], and modulations of the muscle proteome [13,14] and transcriptome [15]. However, muscle mass varies over a lifetime; it reaches maximal levels in middle-aged adults (up to 40 years of age) and progressively declines then after [16]. Regular exercise is a primary preventive approach against age-related muscle wasting [4]. There is an increased participation of master athletes (i.e., > 40 years old) in endurance and ultra-endurance (lasting more than 6 h) events [17]. Given their level of performance, master athletes represent a model of successful aging. Aging, nonetheless, results in a decrease in endurance performance. Indeed, for both non-elite and elite endurance athletes, peak endurance performance is maintained in middle-aged adults until 35 years of age but is followed by a modest decrease for master runners until 50–60 years of age, with progressively steeper declines thereafter [18].

Despite no evidence for major differences in performance and in long-term muscle remodeling between middle-aged and master endurance athletes, we hypothesized that the short-term plasticity of the skeletal muscle might be altered for master athletes when they are challenged by a single bout of prolonged aerobic exercise. The present study focuses on 24-h treadmill ultra-endurance running (24TR), and we assessed if muscle proteome, fiber morphometry, intramyocellular LD, oxidative activity, ECM and micro-vascularisation differ between master and middle-aged runners.

## 2. Results

### 2.1. Subject Clinical Characteristics

Table 1 summarizes the main clinical and physiological characteristics of the subjects involved in the present study. Body weight, body mass index, ultra-endurance experience, training volume and 24TR performance were similar between middle-aged (37Y) and master (50Y) athletes. The physiological variables, VO_2max_, velocity at VO_2max_, average speed sustained over the 24TR, velocity associated with the lactate inflection point (V_4mmol_), and running economy (RE) at 8 km/h also did not differ between 37Y and 50Y runners.

### 2.2. No Age-Related Effect of 24TR on Fiber Type Distribution

The human skeletal muscles are of mixed fiber-type composition, as they comprise slow-oxidative (type-I), fast-oxidative-glycolytic (type-IIA), and fast-glycolytic (type-IIX), together with hybrid fibers. Myosin heavy chain specific antibodies were used to assess contractile types in the human biopsies (Figure 1A), and on average, 232 (114–397) fibers per individual pre-24TR (PRE) and post-24TR (POST) were analyzed for contractile type, cross-sectional area (CSA), perimeter and shape. As shown in Figure 1B, 37Y or 50Y runners exhibited similar PRE fiber type distribution. No difference in fiber type distribution was observed between PRE or POST muscles for either 37Y or 50Y runners. For all runners, type-I and then type-IIA fibers were the most abundant, while type-IIX and hybrid fibers (type I-IIA and type IIA-IIX) were scarce and not further investigated.

### 2.3. Age-Related Effect of 24TR on Fiber Type Morphometry

While the proportion of the different fiber types remained constant, the mean fiber cross-sectional area (CSA) changed in response to 24TR, though only for 50Y subjects. As shown in Figure 1C, for PRE-24TR, the mean CSA of type-I and type-IIA fibers were similar between 37Y and 50Y runners, and they remained similar POST for 37Y subjects. However, for 50Y subjects, type-I CSA tended to, and type-IIA CSA did decrease in response to 24TR. A frequency histogram of type-I and type-IIA fiber CSA confirmed that for 50Y athletes, 24TR was associated with a shift to fibers with a smaller CSA (Figure 1D). The reduction in fiber CSA could be associated with an altered shape, such as the flattening of fibers. To assess this point, the perimeter of each fiber was measured to calculate a shape factor. As shown in the Appendix A, no change in type-I or type-IIA fiber perimeter (Appendix A) and shape (Appendix A) could be detected in response to 24TR for either 37Y or 50Y runners.

### 2.4. Age-Related Effect of 24TR on Mitochondrial Enzymatic Activities

In addition to morphological and contractile properties of muscle fibers, 24TR may affect intramyocellular organelles, such as mitochondria and lipid droplets. Succinate dehydrogenase (SDH) and cytochrome c oxidase (COX) are classically used to assess mitochondrial citric cycle and oxidative activities, respectively. For each runner PRE and POST, an average of 220 (125–308) fibers were analyzed for SDH and COX activity. As shown in Appendix A, COX activity remained unaffected by 24TR in types I and IIA fibers in both 37Y and 50Y runners. However, SDH activity decreased after 24TR in both types I and IIA fibers in 50Y athletes (Appendix A).

### 2.5. Age-Related Effect of 24TR on Muscle Lipid Droplets

Oil red O staining of neutral lipids was used to assess intramyocellular lipid droplets (LDs) and to calculate a fiber type-specific lipid content index (LI) (Figure 2A). PRE-24TR, intramyocellular lipids accumulated more in slow-oxidative fibers, as LI was 3–4 times higher in type-I than in type-IIA fibers. There was no group difference before 24TR, as fiber type-I and type-IIA specific LIs were similar between 37Y and 50Y runners (Figure 2B). However, 24TR sharply affected intramyocellular lipids and this in an age-related fashion. For 37Y runners, 24TR strongly decreased type-I fiber LI, but not type-IIA fiber LI. In contrast, for 50Y runners, 24TR poorly altered muscle lipid droplets, as there was only a trend for 24TR to decrease type-I fiber LI (*p* = 0.097).

Modifications in fiber-specific LI could be due to changes in LD number and/or LD area. Image analysis was then used to assess 20,000–60,000 LDs per subject. These data indicated that 24TR decreased the LD number in type-I fibers for both 37Y and 50Y runners (Figure 2C left). However, this decrease in LD number was twice as important for 37Y than for 50Y runners (89% vs. 37%, respectively). No 24TR-dependent change in the LD number (Figure 2C right) or mean-area (not shown) was observed for type-IIA fibers. The frequency histogram of the area of all individual LDs in type-I fibers indicated that 24TR was associated with a shift to smaller droplets for 37Y, but not for 50Y runners (Figure 2D).

### 2.6. Age-Related Effect of 24TR on Muscle Extracellular Matrix

Besides muscle fibers, the functional assembly of the skeletal muscle is governed by the extracellular matrix (ECM) [19]. ECM was then investigated by using Sirius red that labels major ECM constituents (collagens I and III) [20] and using image analysis to distinguish the ECM endomysium from the perimysium (Figure 3A). The ECM is critical to maintaining structures and to transfer forces, and interestingly, the regression analysis that included all PRE-24TR subjects indicated that the endomysium area was negatively correlated with running economy (RE) (*r* = 0.87, *p* = 0.004). As shown in Figure 3B, 24TR increased the endomysium’s thickness for 37Y but not for 50Y runners.

### 2.7. Age-Related Effect of 24TR on Muscle Capillarization

The ECM contains various stromal cells, including stem cells, immune cells, adipocytes and capillaries. The remodeling of the ECM that occurs with 24TR for 37Y runners might be associated with alterations in capillarization. To investigate this point, we further assessed blood capillaries using an anti-CD31 that recognizes a trans-membranous glycoprotein (PECAM-1) specifically expressed by vascular endothelial cells [21]. On average, 232 (153–307) capillaries per participant were analyzed PRE- and POST-24TR (Figure 4A). As shown in Appendix A, 37Y and 50Y runners exhibited similar capillarization indexes PRE-24TR. Moreover, 24TR did not modify the capillary density (CD) and fiber type-specific indices, such as capillary-to-fiber ratio (CAF), individual capillary-to-fiber ratio (C/Fi), and capillary-to-fiber perimeter (CFPE) in 37Y and 50Y muscles.

However, 24TR did change the functional surface of exchange between capillaries and muscle fibers (LC/PF), and this is an age-related fashion. Specifically, for 37Y runners, LC/PF significantly increased POST compared to PRE, while no significant change occurred for 50Y runners. Moreover, increases in LC/PF of 37Y runners were not dependent on the type of muscle fiber around which capillaries were observed, as it occurred for both type I and IIA fibers (Figure 4B). The increase in LC/PF in the POST group of 37Y runners is mostly due to an increase in the total length of contact (LC) since PF was unchanged (Appendix A). The frequency histogram of the length of contact for each capillary (LCi) confirmed that for 37Y runners, 24TR was associated with a shift to capillaries with longer capillary-to-fiber contacts for both type-I and type-IIA fibers (Figure 4C).

### 2.8. Global Indexes of Oxidative Metabolism, Lipid Droplets and Capillarization

Further studies were then performed to assess coordinated regulations between capillarization, intramyocellular lipids, and mitochondrial energy metabolisms at a fiber type level. As our studies were performed with serial cross-sections, many fibers could be matched between analyses, and for each fiber, we computed global indices of capillarization (LC, the total length of contact with capillaries), intramyocellular lipids (∑LD, the sum of LD area) and mitochondrial oxidative capacity (Int-SDH, the spatially integrated SDH activity), and assessed fiber-to-fiber inter-correlations.

As shown in Figure 5A,C, for both PRE-24TR age groups, all three indices were inter-related for type-I fibers, indicating substantial coordination between vascular, lipid, and mitochondrial processes. However, correlations with the lipid index were less preserved in type-IIA fibers that rely less on lipid metabolism.

Most correlations vanished POST-24TR, suggesting a loss in coordination between vascular, lipid, and mitochondrial processes (Figure 5B). Of note, 37Y appeared more resistant than 50Y muscles, as they maintained correlations between vascular and mitochondrial indexes.

### 2.9. Age-Related Effect of 24TR on the Muscle Proteome

Label-free quantitative protein profiling was finally used to compare the muscle proteomes of 37Y and 50Y runners PRE- and POST-24TR. A total number of 626 proteins were identified and quantitation analyses were performed for proteins detected in all samples. PRE-24TR, 419 proteins were compared between 37Y and 50Y athletes, and only 4 proteins were differentially expressed and downregulated in 50Y muscles (Appendix A): cytoplasmic isoform 3 of malate dehydrogenase (MDH1) is important for the NADH/NAD+ shuttle between cytosol and mitochondria; 4-trimethylaminobutyraldehyde dehydrogenase (ALDH9A1) detoxifies reactive aldehydes; protein FAM162A is involved in the regulation of apoptosis; DCI protein is central for beta-oxidation of unsaturated fatty acids.

A total of 401 and 429 proteins were compared between PRE and POST for 37Y and 50Y runners, respectively. Under our stringent statistical conditions, only two proteins were differentially expressed for 37Y with 24TR (Appendix A): the cAMP-dependent protein kinase type I-alpha regulatory subunit (PRKAR1A) decreased, while the uncharacterized protein C1orf94 increased with 24TR. No protein was differentially expressed in response to 24TR for 50Y runners.

Although significant changes in levels of individual proteins are important, minor but concordant changes in sets of functionally related proteins are also strongly biologically relevant [22]. Instead of testing the differential expression of individual proteins, we then focused on the differential expression of pathway-based sets of proteins and conducted functional class scoring (FCS) using GeneTrail2 [23]. FCS did not identify an altered pathway, neither PRE-24TR between 37Y and 50Y nor in response to 24TR for 37Y runners. However, FCS analyses revealed that 24TR for 50Y runners was associated with reduced integrin and oxidative phosphorylation pathways, which was counterbalanced by an enriched glycolysis (Appendix A and Figure 6).

## 3. Discussion

Endurance exercise induces neuromuscular fatigue, which has both central (i.e., nervous) and peripheral (i.e., skeletal muscle) components. Previous experiments investigating 24TR indicated that the central nervous system is mainly responsible for neuromuscular fatigue [24]. Such a central process could represent a protective mechanism, preventing extensive muscle damage and homeostasis disturbance. Although the muscle peripheral component is not dramatically impaired, a single bout of ultra-endurance exercise is still associated with transient changes in myocellular processes, reflecting metabolic and functional adaptations that enable optimal performance while maintaining homeostasis.

In the present study, we provide evidence that before exercise, muscle fiber type-specific morphometry, intramyocellular LD, mitochondrial oxidative capacity, ECM and micro-vascularisation are similar between master and middle-aged runners. However, despite similar performance and muscle characteristics, we also demonstrate that with exercise, the short-term plasticity of skeletal muscle differs between the two age groups. For master athletes, such “early signs” of muscle aging are revealed by divergences in the transient malleability of the extracellular matrix, micro-vascularisation, intramyocellular lipid droplets, and mitochondrial oxidative capacity in response to a single bout of 24TR.

### 3.1. Short-Term Plasticity of Skeletal Muscle in Middle-Aged Runners

It is often underestimated that muscle physiology depends not only on muscle fibers but also on the extracellular matrix embedding the fibers. The functional assembly of the skeletal muscle is governed by the ECM, which plays a central role in the transmission of contractile force. In fact, most changes in the functional and morphological characteristics of the muscle require a remodeling of the extracellular matrix [19,25]. Previous studies in humans indicated that resistance exercise increases collagen synthesis [26], and endurance training increases prolyl hydroxylase that stabilizes the collagen structure [27]. To our knowledge, however, no data describe ECM changes pre-post endurance exercise, and our present results do indicate that 24TR is associated with endomysium thickening, although this solely occurred for middle-aged (37Y) runners. Whether ECM thickening is related to marked dehydration is unlikely, as our previous data indicated that 24TR is a relatively low-intensity exercise that does not involve ionic imbalance or perturbation in membrane excitability [24,28].

Previous studies indicated that endurance exercise also activates matrix metalloproteinases that have well-documented roles in ECM turnover, the release of growth factors, and angiogenesis [25]. The ECM network provides mechanical support to blood vessels, and the remodeling of the extracellular matrix may be associated with some change in capillarization. Interestingly, in the present study, although 24TR did not affect the indices of capillarization (CD, CAF, CFPE), a single bout of 24TR did enhance the surface of exchange between the capillaries and muscle fibers (LC/PF), and once again, specifically for 37Y runners. Therefore, for middle-aged but not master athletes, endomysium thickening was associated with a greater diffusion capacity between the capillary network and the muscle fibers. Furthermore, it is well-known that endurance athletes develop a higher muscle capillary density than sedentary people [29], and we did observe smaller CD in sedentary young adults [10]. However, to the best of our knowledge, this is the first report describing capillary plasticity in response to a single bout of exercise. The blood flow to the active muscles increases several-fold upon exercise [30], and a higher capillary LC/PF index may then favor O_2_ supply, substrates delivery to, and oxidative metabolism in myofibers. Notably, for 37Y runners, the ECM thickening and increased capillary LC/PF index were indeed concomitant with enhanced intramyocellular lipid metabolism. Whether mechanisms such as intussusceptive angiogenesis [31] are involved in capillary remodeling remains to be specified.

A well-known adaptive change of the skeletal muscle to long-term endurance training is an increase in intramyocellular lipids [32]. Herein, lipid contents (LI) in type-I fibers were twice those we previously measured for untrained individuals [9]. This is in agreement with previous electron microscopy studies showing high volume percentages of lipid in highly-trained individuals [33]. In addition to this long-term training effect, there is an acute “pre-post” effect of a single bout of endurance exercise, as intramyocellular lipids can play a central role as an energy source. Previous electron microscopy [33] and proton magnetic resonance spectroscopy [34] studies indeed showed that intramyocellular, but not extramyocellular, lipids decrease after the completion of a marathon.

Our quantitative analyses of intramyocellular lipids revealed a large decrease in LI immediately post 24TR for 37Y runners. Moreover, our data indicate that this 89% depletion is specific to type-I fibers. Fewer reductions (62%), predominantly in type-I fibers, were previously observed for young adults and shorter-duration (2 h cycling) moderate-intensity exercise [7]. Here, although 24TR could nearly deplete LD reserves in type-I fibers, such an extreme exercise was not able to alter LD in type IIA fibers. This supports the contention that muscle fiber-type recruitment during endurance exercise mainly relies on the use of type I fibers [35]. This remarkable plasticity of lipids droplets further suggests an efficient lipid beta-oxidation in 37Y muscles. Accordingly, our data (SDH, COX, proteome) did not detect major alterations in mitochondrial oxidative capacity for middle-aged runners in response to 24TR.

### 3.2. Short-Term Plasticity of Skeletal Muscle in Master Runners

While the effects of exercise on muscle plasticity have mostly been studied in young or middle-aged populations, very few studies have been conducted in master individuals. The gradual loss of skeletal muscle mass and functionality (e.g., sarcopenia) is a consistent hallmark of aging that nonetheless begins early in a lifetime [16,36]. Muscle mass and strength reach maximal levels in young adulthood (<40 years of age) and start to decline then after. Even for elite athletes, peak endurance performance is reported to decrease for middle-aged men after 35 years of age [18,37]. In the present study, middle-aged (37Y) and master (50Y) athletes exhibited similar physiological parameters for ultra-endurance (VO_2max_, V_VO2max_, V_4mmol_, RE). Before 24TR exercise, there was also no detectable age-related modification in muscle characteristics, and both 37Y and 50Y runners exhibited similarly high levels of capillary density, intramyocellular lipids, and oxidative capacity, compared to untrained individuals [9,10]. Moreover, our proteomic analysis did not reveal major differences between middle-aged and master runners, as only 4 out of 429 proteins were differentially expressed between the 2 age groups before 24TR; 2 proteins, however, pointed to potential alterations in energy metabolism with reduced NADH/NAD^+^ shuttle (MDH1) and beta-oxidation (DCI) for master runners.

Despite similar physiological and muscle characteristics, master athletes still differ from middle-aged ones with regard to muscle plasticity when they were challenged by a single bout of 24TR. In contrast to middle-aged runners, master athletes exhibit some shrinkage of muscle fibers. Similar findings, so far unexplained, were previously reported after a long-distance skiing race [38] or an extremely long-distance run [39]. Dehydration readily occurs in marathon runners [40] and muscle fibers could be changing in size due to fluid shifts. However, 24TR is a relatively low-intensity exercise, and our previous data did not provide evidence for marked dehydration, such as an ionic imbalance or perturbation in membrane excitability [24,28]. Whether alterations in energy metabolism partly account for fiber shrinkage remains to be established.

Moreover, in strict contrast to middle-aged runners, master athletes did not reveal pre-post plasticity in the ECM endomysium and capillary LC/PF index. Likewise, intramyocellular lipid depletion was strongly age-dependent and appeared limited for 50Y compared to 37Y runners. While 37Y runners exhibited reductions in both the number and size of lipid droplets, 50Y master runners exhibited only fewer lipid droplets of similar size after running. The few studies previously investigating the effects of moderate-intensity exercise were of short duration (1–2 h), however, they similarly reported that intramuscular lipids form an important substrate source for young (23 years old) [7], but not for older (59 years old) athletes [41]. An impaired mitochondrial oxidative metabolism could partly account for a limited lipid depletion in master muscles. In agreement with this hypothesis, the 24TR ultra trail was indeed associated with decreased SDH activity, and functional class scoring of the proteomes revealed reduced oxidative phosphorylation exclusively for 50Y runners.

The main limitation of our study is that we could only include a low number of athletes. Further works are also required to investigate the influence of gender. More experimental studies are also necessary to perform 3D analyses of capillarization and to characterize the intramyocellular localization and composition of the lipid droplets. We believe that the present paper is an important contribution, but it must be seen as a first step toward the identification of early signs of muscle aging.

## 4. Materials and Methods

### 4.1. Subject Clinical Characteristics

Six middle-aged (37Y) and five older (50Y) master men were recruited among experienced ultra-endurance runners and all of them had run at least a race longer than 24 h. They had more than 15 years of training history in running and reported to run an average of 81 ± 12 km/week. The protocol was approved by the local ethics committee (Comité de Protection des Personnes Sud-Est 1, France) and registered in http://clinicaltrial.gov (accessed on 10 February 2022) (# NCT 00428779). The study was conducted in accordance with the principles of the revised Declaration of Helsinki, and all subjects provided written informed consent.

### 4.2. 24 TR Procedure

The biopsies used in the present study were acquired in a protocol that was previously described [24,28]. The subjects visited the laboratory twice, with each session separated by 3–4 weeks. In the first session, the subjects performed tests on a motorized treadmill (Gymrol S2500, HEF Tecmachine, Andrezieux, France) to determine the anaerobic threshold (VO_2max)_ and velocities associated with VO_2max_ (V_VO2max_) and with the blood lactate inflection point (V_4mmol_).

The second session consisted of the 24-h treadmill ultra-endurance running protocol (24TR). Subjects were asked to refrain from strenuous exercise for a week before the 24TR. On the day of the experiment, all subjects ate the same lunch at noon. About 2 h before starting the 24TR, a muscle biopsy (~120 mg) was collected under local anesthesia from the superficial portion of the left *vastus lateralis* muscle using a percutaneous technique [42]. A large fascicle of fibers was oriented and included in the embedding medium (Cryomount; Histolab, Gothenburg, Sweden), frozen in isopentane cooled to its freezing point in liquid N_2_, and stored at −80 °C.

After the biopsy, the subjects rested for approximately 2 h and then were asked to start the 24TR. Ten minutes after the start, subjects were asked to run for 4 min at 8 km/h for measurements of running economy (RE). Food and water intakes during the 24TR were managed by the subjects as in a normal race. Post-exercise, a second biopsy sample (~120 mg) was collected from the superficial portion of the right *vastus lateralis* within 10 min following the end of the run.

### 4.3. Fiber Type and Morphometry

Serial muscle cross-sections (10 µm thick) were obtained using a cryostat (Microm, Francheville, France) at −25 °C. Two serial cross-sections were labeled with monoclonal antibodies against myosin heavy-chain-I (MHC-I) (A4.951 from Enzo Life Sciences, Villeurbanne, France) or MHC-IIa (N2.261, Enzo), and co-labeled with anti-laminin-α1 (Sigma, Saint-Quentin-Fallavier, France) to outline the fibers, and resolved with corresponding secondary antibodies conjugated to Alexa Fluor 488 or 546 (Invitrogen, Cergy-Pontoise, France). Images were captured with a high-resolution cooled digital DP-72 camera coupled to a BX-51 microscope (Olympus, Rungis, France). The contractile type (I, I-IIA, IIA, IIA-IIX, or IIX), cross-sectional area (CSA), and perimeter (PF) were determined for each fiber using Visilog-6.9 (Noesis, Gif-sur-Yvette, France) as previously described [43]. A shape factor (PF^2^/4π/CSA) was calculated, a value of 1.0 indicated a circle, and a value > 1.0 indicated an increasingly elongated ellipse.

### 4.4. Mitochondrial Histoenzymology

Cytochrome c oxidase (COX) and succinate dehydrogenase (SDH) activities were determined histochemically on serial cross-sections as previously described [44]. COX and SDH optical densities were quantified for each fiber and COX, SDH and MHC images were matched using Visilog-6.9 software. Spatially integrated SDH activity (Int-SDH) was calculated as the product of SDH activity and CSA for each fiber [45]. Int-SDH is related to the total volume of mitochondria and to the maximum oxygen uptake rate of a fiber (VO_2max_._fiber_) [45,46], and a regression analysis including all subjects PRE-24TR confirmed that the overall mean of Int-SDH is positively correlated with whole-body VO_2max_ (*r* = 0.83, *p* = 0.021).

### 4.5. Intramyocellular Lipid Droplets

Oil red O (ORO, Sigma) stock solution (500 mg/mL ORO in 60% triethylphosphate in water (*v*/*v*)) was diluted with 0.67 vol water and filtered before use. Cross-sections were air-dried, incubated in 4% paraformaldehyde (*v*/*v*) for 30 min, washed thrice with PBS, labeled with anti-laminin-α1, and incubated with a secondary antibody conjugated to Alexa Fluor 488. Cross-sections were then incubated with ORO diluted solution for 20 min and washed thrice with water. Pictures were rapidly captured and saved as gray-scale images. ORO and MHC images were matched and analyzed with ImageJ (http://rsb.info.nih.gov/ij/) (accessed on 10 February 2022) to determine for each fiber the number and area of lipid droplets (LD) and a lipid content index (LI, i.e., the percentage fiber area occupied by LD), as previously described [9].

### 4.6. Extracellular Matrix

For the identification of extracellular matrix (ECM) perimysium and endomysium, cross-sections were fixed for 1 h in 100% acetone and stained for 10 min with Picro-formalin solution, containing 7.5% picric acid and 4.5% formaldehyde (Sigma) in 95% ethanol. After washing in 90% ethanol (1 min) and deionized water (10 min), cross-sections were stained for 1 h with 0.1% picrosirius red, incubated for 5 min in 0.01 M HCl solution, and then dehydrated and mounted with Safe Mount (Labonord, Templemars, France) as previously described [21]. Image acquisitions were obtained in the bright-field mode, and image analyses were performed using Visilog-6.9 software. The green component of the initial image was used for higher contrast. Top-hat filtering, followed by manual thresholding on the gray level, allowed segmentation of the connective tissue network (perimysium and endomysium).

### 4.7. Capillary Network

Cross-sections were labeled with anti-CD31 (M0823 from Dako, Glostrup, Denmark), co-labeled with anti-laminin-α1, and resolved with corresponding Alexa Fluor secondary antibodies as previously described [10]. Capillary and MHC images were matched using Visilog-6.9, and the length of contact and tortuosity were measured using ImageJ. A mean of 450 fibers and 800 capillaries were analyzed per subject.

Capillary density (CD) was expressed as the number of capillaries counted per square mm. Individual fiber indices were determined as previously described [47]. Briefly, for each fiber type, the mean number of capillaries around a single fiber (CAF or CC) was calculated. The capillary-to-fiber ratio (C/Fi) taking into account the sharing factor, and the capillary-to-fiber perimeter exchange index (CFPE) were calculated according to [48]. The capillary-to-fiber interface was estimated using the LC/PF index, which is the total length of contact (LC) between all capillaries and the fiber divided by the perimeter of the same fiber (PF).

### 4.8. Proteomics and Bioinformatics

Label-free quantitative protein profiling was used to analyze each muscle biopsy and is described in the supplementary methods. The mass spectrometry proteomics data have been deposited with the ProteomeXchange Consortium [49] via the PRIDE partner repository with the dataset identifier PXD006840. The protein–protein interaction network was investigated using the Search Tool for Retrieval of Interacting Genes (STRING, version 11.0) database (https://string-db.org) (accessed on 10 February 2022) [50]. STRING analysis options were based on evidence mode, highest confidence (0.9) and we used MCL clustering and 1.8 inflation to reveal sub-grouping within the networks. Functional class scoring of pathway-based sets of proteins (FCS) was performed using the GeneTrail2 web service [23] and a Kolmogorov–Smirnov test with Benjamini–Yekutieli correction (*p* < 0.01) for multiple testing.

### 4.9. Statistical Analyses

Unless otherwise stated, data are presented as a means ± standard error of the mean (SEM). The differences between groups were assessed using repeated-measures ANOVA, followed by a post hoc Fisher’s test for pairwise comparisons between groups. For label-free quantitative protein profiling, paired (PRE vs. POST) or unpaired (37Y vs. 50Y) Student’s t-tests were followed by a correction for multiple testing according to [51]. Univariate linear Pearson’s regressions were carried out to investigate the relationships between variables. Statistical analyses were performed using XLSTAT (Addinsoft, Paris, France) and the significance was set at *p* < 0.05.

## 5. Conclusions

Collectively, our data support the idea that middle-aged and master endurance athletes exhibit similar physiologic characteristics and long-term adaptations of the skeletal muscle. Nevertheless, they reveal distinct transient plasticity to a single bout of ultra-endurance exercise. While reflecting short-term adaptations to maintain homeostasis, such distinct malleability of master athlete muscles may constitute early signs of muscle aging.

## Figures and Tables

**Figure 1 ijms-23-03713-f001:**
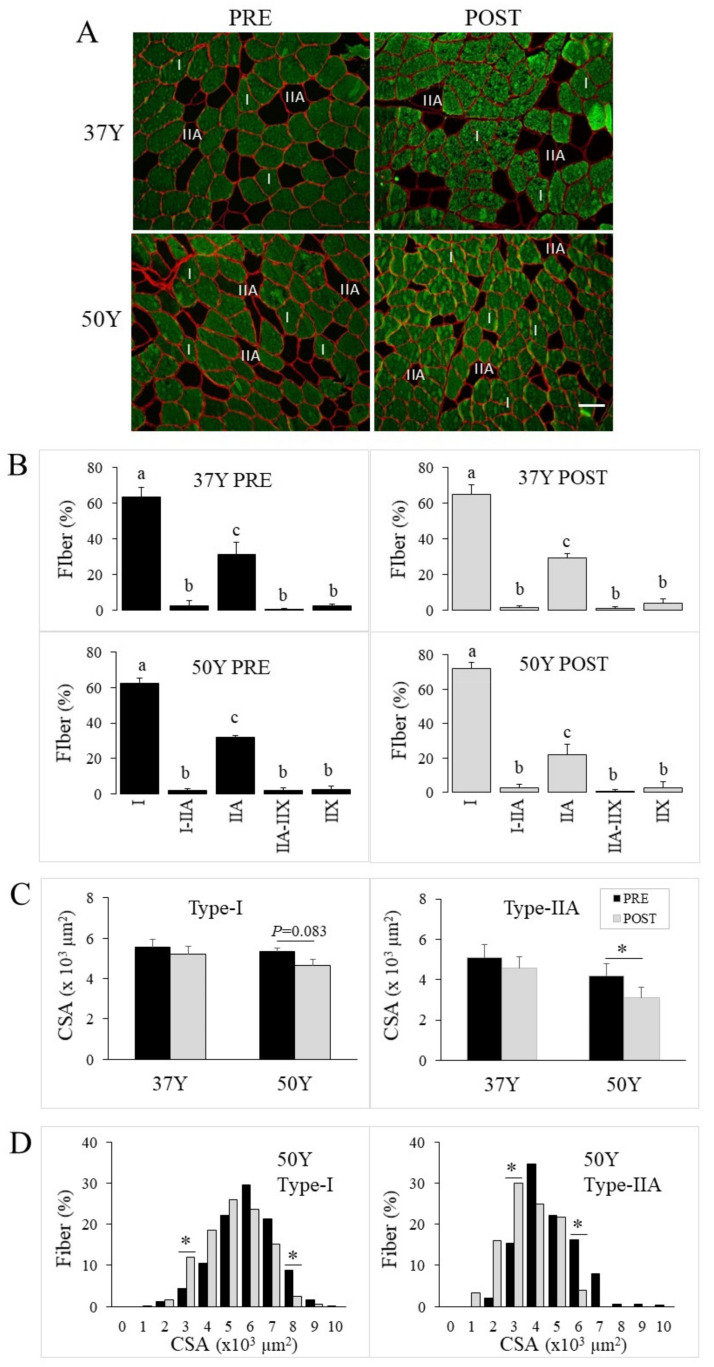
Fiber type-specific distribution and morphometry. *Vastus lateralis* biopsies were from 37Y and 50Y runners before (PRE) and after (POST) 24TR. (**A**) Representative images of cross-sections labeled for myosin heavy chain I (green) and counter-stained for laminin-α1 (red) to outline the fibers; the scale bar represents 50 µm. (**B**) The fiber type proportion was measured PRE (black bars) and POST (gray bars) 24TR for middle-aged (37Y) and master (50Y) runners. Different letters indicate a significant difference (*p* < 0.05) between fiber types. (**C**) Mean cross-section area (CSA) of type-I (left) and type-IIA (right) fibers for 37Y and 50Y runners PRE- and POST-24TR; statistical interaction was *p* = 0.049 between time and group for type-IIA CSA. (**D**) Frequency histogram of CSA for type-I (left) and type-IIA (right) fibers PRE- and POST-24TR for 50Y runners. * *p* < 0.05 between PRE and POST.

**Figure 2 ijms-23-03713-f002:**
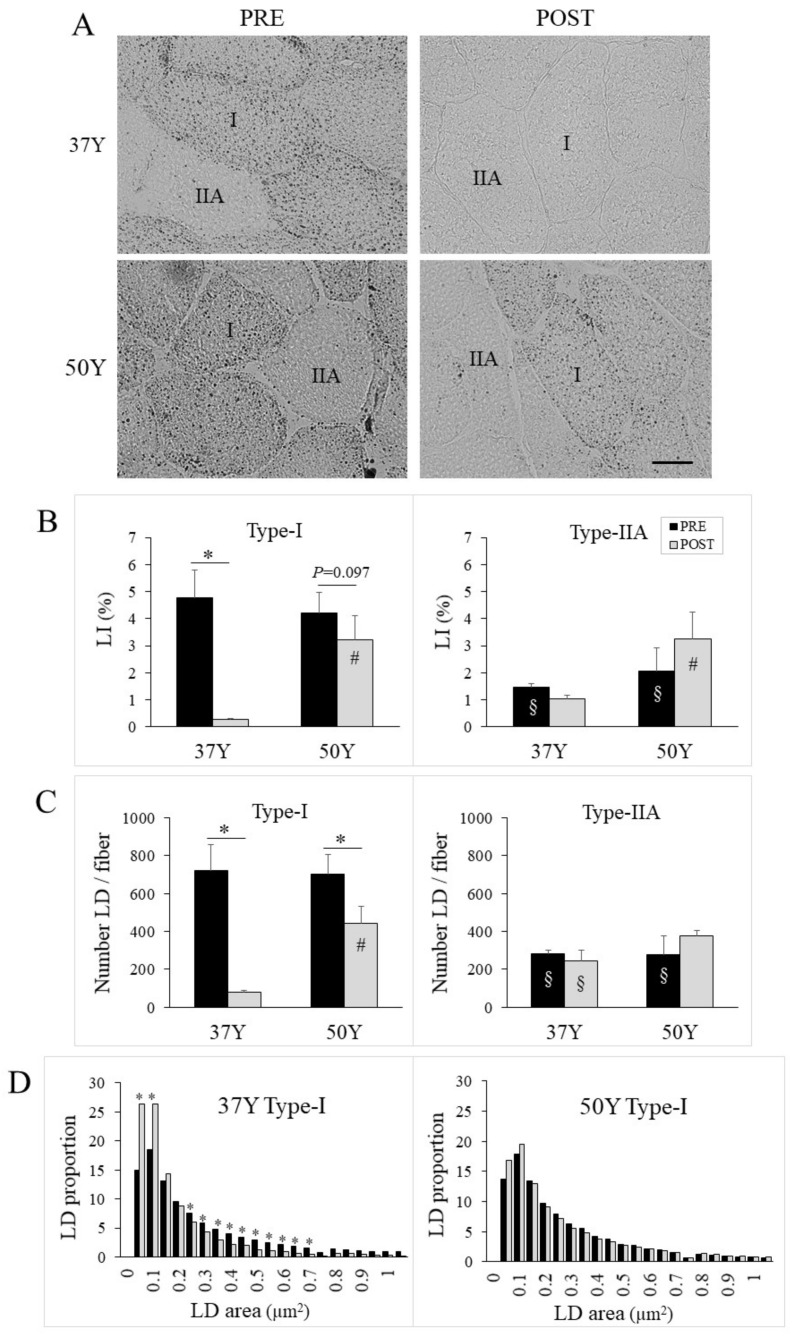
Intramyocellular lipid content. (**A**) Representative images of cross-sections stained with Oil red O for PRE and POST muscles of 37Y and 50Y runners; the scale bar represents 20 µm. (**B**) Lipid content index (LI) expressed as a percentage of fiber area occupied by lipid droplets (LD); statistical interaction was *p* = 0.037 between time and group for type-I fiber. (**C**) Mean number of LD per fiber in type-I (left) and type-IIA fiber (right) for PRE (black bars) and POST (gray bars) muscles of middle-aged (37Y) and master (50Y) runners; statistical interaction was *p* = 0.045 between time and group for type-I fiber. (**D**) Frequency histogram of droplet area for type-I fibers in PRE (black bars) and POST (gray bars) muscles of 37Y (left) and 50Y (right) runners. * *p* < 0.05 between PRE and POST; # *p* < 0.05 between 37Y and 50Y runners; § *p* < 0.05 between type-I and type-IIA fibers.

**Figure 3 ijms-23-03713-f003:**
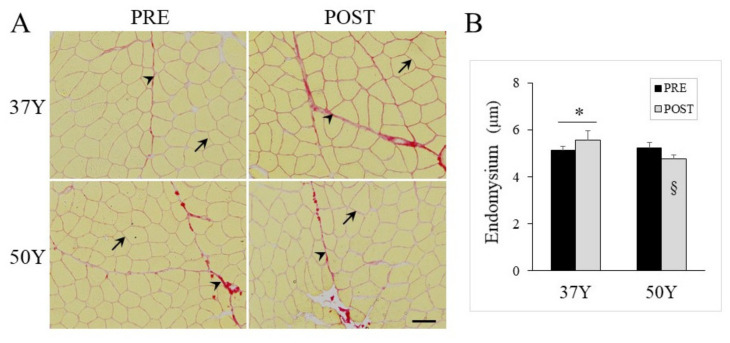
Muscle extracellular matrix (ECM) endomysium. (**A**) Representative images of cross-sections stained with Sirius red indicating endomysium (arrow) and perimysium (arrowhead) for middle-aged (37Y) and master (50Y) runners PRE- and POST-24TR; the scale bar represents 50 µm. (**B**) ECM endomysium mean thickness for 37Y and 50Y runners PRE (black bars) and POST (gray bars) 24TR.). Statistical interaction was *p* = 0.031 between time and group. § *p* < 0.05 between type-I and type-IIA fibers; * *p* < 0.05 between PRE and POST muscles.

**Figure 4 ijms-23-03713-f004:**
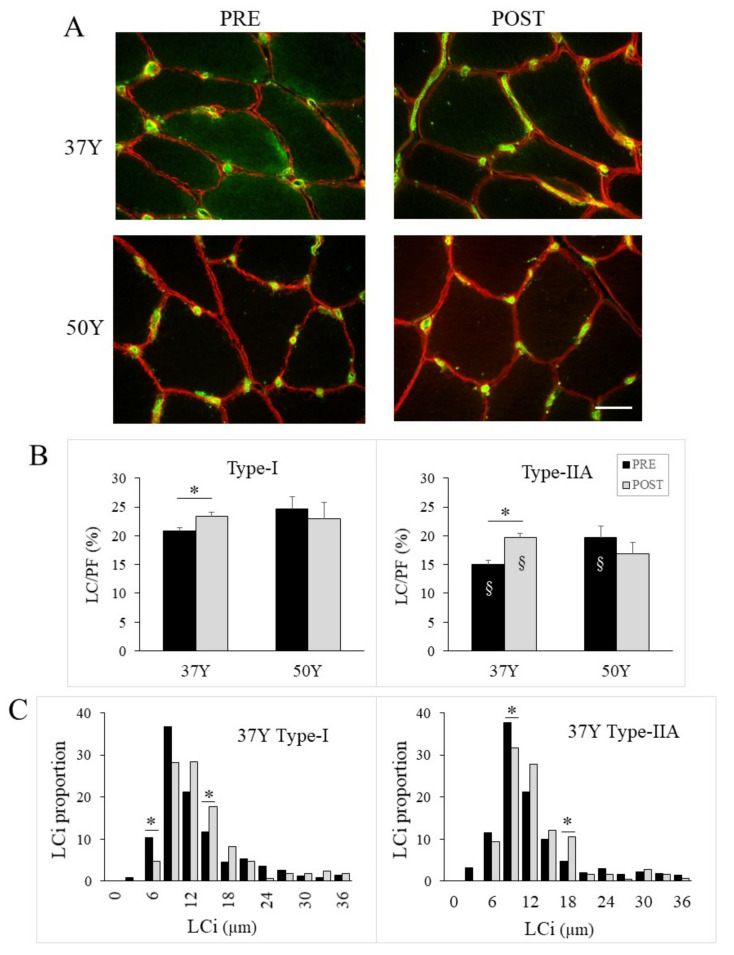
Muscle capillarization. (**A**) Representative images of cross-sections labeled with anti-CD31 antibody for 37Y and 50Y runners PRE- and POST-24TR; the scale bar represents 20 µm. (**B**) The functional surface of exchange between capillaries and muscle fibers (LC/PF) is expressed as a percentage of fiber perimeter in contact with capillaries for middle-aged (37Y) and master (50Y) runners PRE (black bars) and POST (gray bars) 24TR. Statistical interaction between time and group was *p* = 0.001 for type-I and *p* = 0.003 for type IIA fibers. (**C**) Frequency histogram of the length of contact of each capillary (LCi) for type-I (left) and type-IIA fibers (right) in PRE (black) and POST (gray) muscles of 37Y runners. * *p* < 0.05 between PRE and POST.

**Figure 5 ijms-23-03713-f005:**
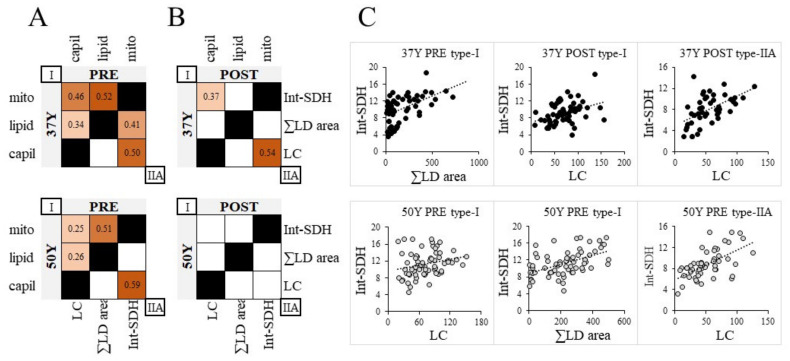
Coordination between microvascular, lipid, and mitochondrial processes. Linear Pearson’s correlations between global indices of capillarization (LC, the total length of contact with capillaries), intramyocellular lipids (∑LD area, the sum of LD area), and mitochondrial oxidative capacity (Int-SDH) for type I (upper left) and type IIA (lower right) fibers of (**A**) PRE and (**B**) POST vastus lateralis of 37Y (top) and 50Y (bottom) runners. Correlations coefficients (*r*) are indicated when *p* < 0.05. (**C**) Examples of linear regressions between Int-SDH, ∑LD area, and LC for 37Y (top) and 50Y (bottom) runners.

**Figure 6 ijms-23-03713-f006:**
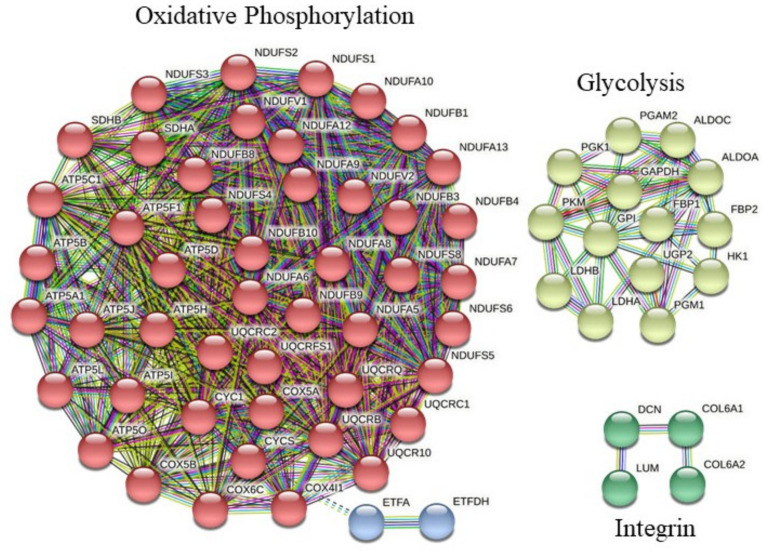
Pathways affected by 24TR in 50Y muscles: protein–protein interaction network. The interaction map was generated using STRING with the highest confidence (0.90), MCL clustering (1.8 inflation parameter), and all criteria for linkage (co-occurrence, co-expression, experiments, neighborhood, databases, text-mining, and homology).

**Table 1 ijms-23-03713-t001:** Subjects characteristics.

	37Y (*n* = 6)	50Y (*n* = 5)
Age (yr)	37.0 ± 0.4	50.3 ± 2.3 *
Body weight (kg)	72.3 ± 2.4	76.3 ± 5.2
BMI (kg/m^2^) ^1^	22.9 ± 0.6	24.6 ± 1.4
Ultra-endurance experience (yr)	7.5 ± 2.0	6.5 ± 1.4
Training volume (km/week)	87.5 ± 6.7	73.6 ± 3.8
24TR Performance (km)	150.5 ± 5.1	139.3 ± 8.6
Effective running time (h)	20.5 ± 0.6	21.4 ± 0.6
VO_2max_ (mL/min/kg)	53.4 ± 2.5	49.3 ± 3.5
V_VO2max_ (km/h)	18.8 ± 0.6	17.1 ± 0.7
Velocity during running (%V_VO2max_)	39.5 ± 2.2	38.0 ± 0.4
V_4mmol_ (%V_VO2max_)	88.0 ± 1.1	88.0 ± 2.5
RE (mL/min/kg)	28.5 ± 0.9	28.6 ± 0.9

^1^ BMI, body mass index; V_VO2max_, velocity associated with VO_2max_; V_4mmol_, the velocity at blood lactate 4 mM; RE, running economy, i.e., VO_2_ at 8 km/h. Data are presented as means ± SEM. * *p* < 0.05 vs. 37Y.

## Data Availability

The data that support the findings of this study are openly available in the ProteomeXchange Consortium with the dataset identifier PXD006840, and in INRA Dataverse at https://data.inrae.fr/dataset.xhtml?persistentId=doi:10.15454/2DVOZS (accessed on 10 February 2022).

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
