# Peer review of "A Single Bout of Ultra-Endurance Exercise Reveals Early Signs of Muscle Aging in Master Athletes"

_ijms, 2022, doi:10.3390/ijms23073713_

Round 1

Reviewer 1 Report

The study presented here looks at the short-term plasticity of skeletal muscle in elite athletes. The authors postulate that there may be a significant change in skeletal muscle plasticity between age groups when challenged by a single prolonged bout of moderate intensity exercise. Six highly trained middle-aged (37 yrs) and five older (50 yrs) athletes were included in the study.

introduction
The introduction is detailed and outlines the research problem. The literature review could be more detailed here.

Methods
The methods part should follow directly after the introduction.
The sample is very small. How big is the effect/power assumed here, or how big is the actual effect size of the study?

Results

The analysis is very finely structured. 
In Results, the reporting of statistical results is missing, e.g. ANOVA, post-hoc Fisher's test, etc.! Without a detailed presentation of the statistics, an evaluation and classification is not possible for the reader!
Discusion.
the discussion is detailed and deals extensively with the results in the context of previous work. A comparison of the two age groups comes somewhat short in the discussion.

Author Response

We thank the Supervising Editor and the Reviewer’s for their valuable comments regarding our investigation. We have revised the manuscript appropriately and feel is has strengthened this study. Specific responses can be found below which address each individual comment.

Reviewer 1

Comments and Suggestions for Authors

The study presented here looks at the short-term plasticity of skeletal muscle in elite athletes. The authors postulate that there may be a significant change in skeletal muscle plasticity between age groups when challenged by a single prolonged bout of moderate intensity exercise. Six highly trained middle-aged (37 yrs) and five older (50 yrs) athletes were included in the study.

  1. Introduction
    The introduction is detailed and outlines the research problem. The literature review could be more detailed here.

Answer 1.1 – We now provide a more detailed review about muscle sarcopenia (lines 64-70):

“The loss of skeletal mass and function during the aging process (sarcopenia) is one of the most dramatic changes affecting the human body. Most previous studies investigating human sarcopenia relied on the comparison between young (e.g. 20-30 years) and old subjects (65-75 years). They provided major informations about changes in fiber mor-phology, oxidative metabolism, lipid droplets [9], capillarization, ECM fibrosis [10], ions and oxylipins homeostasis [11, 12], and modulations of the muscle proteome [13, 14] and transcriptome [15]. However, muscle …“

  1. Methods     The methods part should follow directly after the introduction.

Answer 1.2 – According to the Instructions for Authors we used ICMS Microsoft Word Template to write the manuscript. In this Word Template, the Materials and Methods section follows the Discussion.

  1. The sample is very small. How big is the effect/power assumed here, or how big is the actual effect size of the study?

Answer 1.3 – The actual PRE-POST effect size (Hedges’g) was calculated for all parameters that were significantly different in ANOVA.

Figure

Age-Group

Fiber

Parameter

Hedges’ g

1C right  

50Y

Type-IIA

CSA

1.09

2B left

37Y

Type-I

LAI

2.73

2C left

37Y

Type-I

LD/fiber

3.27

2C left

50Y

Type-I

LD/fiber

1.52

3B

37Y

ECM thickness

1.45

4B left

37Y

Type-I

LC/PF

2.27

4B right

37Y

Type-IIA

LC/PF

3.90

S1D

50Y

Type-I

SDH

1.41

S1D

50Y

Type-IIA

SDH

2.83

As shown in the table above, all effect sizes were large (>0.80) to huge (>2.00).

  1. Results

The analysis is very finely structured. 
In Results, the reporting of statistical results is missing, e.g. ANOVA, post-hoc Fisher's test, etc.! Without a detailed presentation of the statistics, an evaluation and classification is not possible for the reader!

Answer 1.4 – Because our data correspond to an experiment in which 2 groups (37Y and 50Y) of runners have been followed at 2 different times of a treatment (PRE- and POST-24TR), differences between groups were assessed using repeated measures ANOVA, followed by post-hoc Fisher’s test for pairwise comparisons between groups”. This point is indicated in the Materials and Methods section (§4.9, lines 483-485).

  1. Discusion.

the discussion is detailed and deals extensively with the results in the context of previous work. A comparison of the two age groups comes somewhat short in the discussion.

Answer 1.5 – As indicated in the Discussion, very few studies have been conducted on muscle of master individuals. The effects of exercise on muscle plasticity have mostly been studied in young or middle-aged populations, and to the best of our knowledge, no other study compared muscle properties of middle-aged and master athletes. It was therefore difficult to refer to literature comparing these two age groups.

Reviewer 2 Report

I read with pleasure the results of a your study on muscular changes following 24h treadmill run in a group of 37 year old (middle-aged) and 50 years old (veteran) athletes showing acute changes post exercise in both groups, but demonstrating also age-related differences. It is a well-written, clinically important study showing declining capabilities of human organisms with age (even trained ones). It should be taken as a precaution for older athletes, trainers and physician planning ultra-endurance goals. I have some comments, which in my opinion could improve the manuscript:

  1. A central graphical figure showing main observations in both groups (37Y and 50Y) in the context of 24h run would be useful.
  2. Details regarding 24h run for both studied groups should be provided in the results section (average time and pace, food and water intake, medications if any?)
  3. There is no limitations section – it should be noted that there was no control group per se and the study was only able to analyze the acute changes and not the prolonged ones. Also, the athletes were not highly trained looking at the VO2 max, which did not exceed 60 ml/min/kg in any runner.
  4. I cannot find the supplementary figures/materials – the ones downloaded are the main figures from the text
  5. The manuscript should undergo a sweep for typographical and grammatical errors for example line 38-39, line 121
  6. Please add reference to line 65
  7. Line 108-109 – what do you mean by “Different letters ….” Please explain

Author Response

We thank the Supervising Editor and the Reviewer’s for their valuable comments regarding our investigation. We have revised the manuscript appropriately and feel is has strengthened this study. Specific responses can be found below which address each individual comment.

Reviewer 2

Comments and Suggestions for Authors

I read with pleasure the results of a your study on muscular changes following 24h treadmill run in a group of 37 year old (middle-aged) and 50 years old (veteran) athletes showing acute changes post exercise in both groups, but demonstrating also age-related differences. It is a well-written, clinically important study showing declining capabilities of human organisms with age (even trained ones). It should be taken as a precaution for older athletes, trainers and physician planning ultra-endurance goals. I have some comments, which in my opinion could improve the manuscript:

1. A central graphical figure showing main observations in both groups (37Y and 50Y) in the context of 24h run would be useful.

Answer 2.1 - A graphical abstract summarizing our main observations was submitted to IJMS.

2. Details regarding 24h run for both studied groups should be provided in the results section (average time and pace, food and water intake, medications if any?)

Answer 2.2 - Details regarding the 24h run are described in §2.1 (Subject clinical characteristics) and in Table 1. They include performance (number of km run during 24h), average speed sustained over the 24TR (%VVO2max). The effective running time is now also included in Table 1, and it did not differ between 37Y (20.5 + 0.61 h) and 50Y (21.4 + 0.64 h) runners.

There was no medication for any runner. As indicated in Materials and Methods (§4.2. 24TR Procedure): “Food and water intakes during the 24TR were managed by the subjects as in a normal race”; food and water intakes were not recorded, but they were very similar between runners.

3. There is no limitations section – it should be noted that there was no control group per se and the study was only able to analyze the acute changes and not the prolonged ones.

Answer 2.3a - A limitation section is now included in the discussion (lines 375-380):

“The main limitation of our study is that we could only include a low number of athletes. Further works are also required to investigate the influence of gender. More experimental studies are also necessary to perform 3D analyses of capillarization and to characterize the intramyocellular localization and composition of the lipid droplets. We believe that the present paper is an important contribution, but it must be seen as a first step toward identification of early signs of muscle aging.”

Also, the athletes were not highly trained looking at the VO2 max, which did not exceed 60 ml/min/kg in any runner.

Answer 2.3b - Although VO2max do not exceed 60 ml/min/kg, it is known that VO2max is not as elevated in ultra-endurance runners as it is in shorter distances runners of comparable level. Although VO2max is still important in ultramarathon, it plays a much less important role in performance than in shorter distances. See for instance our recent paper (https://pubmed.ncbi.nlm.nih.gov/35213820/).

In addition, the athletes indicated training volumes > 70 km/week and ultra-endurance experience > 6.5 years. So, they were definitively not elite runners but still can be considered as highly trained. It is not possible to regularly participate in ultra-endurance events without being so.

4. I cannot find the supplementary figures/materials – the ones downloaded are the main figures from the text

Answer 2.4 - We did supply a supplementary zip-file (“Coudy_Supplementary_Files”) during the submission to IJMS. There must have been some problem during MDPI processing of the manuscript.

5. The manuscript should undergo a sweep for typographical and grammatical errors for example line 38-39, line 121

Answer 2.5 – This was modified (line 128): “To assess this point, the perimeter of each fiber was measured to calculate a shape factor”.

6. Please add reference to line 65

Answer 2.6 - A reference (Cruz-Jentoft, 2019) was added (line 72).

7.Line 108-109 – what do you mean by “Different letters ….” Please explain

Answer 2.7 - Fiber types with identical letter (“b”) show identical distribution (e.g. types I-IIA, IIA-IIX and IIX in Fig 1B). However, the distribution of type-I fiber (the only one with “a”) differs from the distribution of all other fiber-types.

Reviewer 3 Report

The authors studied the effects of single bout of prolonged moderate-intensity exercise (24h treadmill run) on proteomics, muscle fibre morphometry, intramyocellular lipid droplet content, mitochondrial oxidative activity, extracellular matrix and capillarisation. The paper is very interesting and has merit. I have a few comments:

  1. The authors studied cross-sectional areas of muscles fibres. CSA is very sensitive to potential oblique sections, which can falsely increase CSA. Since the number of subjects is low, even a few obliquely cut sections could be important confounders. I suggest to also add the data on muscle fibre diameter, especially minimal Ferret diameter which is the least sensitive to the angle of section.
  2. There is thought to be inverse correlation between muscle fibre size and oxidative capacity regardless of the muscle fibre type. It was also shown that large and small fibres can differentially response to pathologic and physiologic conditions (DOI: 1007/s00418-019-01810-7 ). Therefore, the analysis of different parameters (SDH, COX, capillarization, LAI) should be also performed separately for large and small fibres, regardless of the fibre type.
  3. The authors report that the capillarization increased after 24h of exercise. Since 24h is very short time, it is very unlikely that the capillary network would increase so substantially in this time. It is even less likely that the 2D method of studying capillarization, which has low sensitivity and specificity for detecting small changes (DOI: 1016/j.mvr.2009.11.005 ), would be able to show these changes in so few subjects. I believe the most possible explanation is that due to extreme 24h exercise, the muscle was depleted of energy sources and ATP, which could cause that rigor mortis in biopsy sample in a few minutes. In rigor mortis, the muscle fibres contract, and the capillary network around them becomes falsely enlarged.
  4. The authors refer to LC/PF index in abstract and discussion as tortuosity. Since this index is very limited index of tortuosity (note that true tortuosity and branching could be really evaluated only in longitudinal sections or 3D analysis), they should rather refer to it as it is LC/PF index, so that readers that are not expert in 2D capillary analysis or stereology are not misled.
  5. For the same reasons as above, I find it unlikely that the endomysium would actually increase in diameter in 24h due to synthesis of collagen (which is stained by Sirius red). Could the noted increase be the consequence of maybe oedema? Also is there any significant statistical interaction between age and pre and post exercise endomysium thickness (two-way ANOVA)?
  6. Authors use lipid accumulation index (LAI) to measure lipid content in the muscle fibres. Since it does not measure accumulation of lipids, but more of lipid content, the lipid content index would be more appropriate name for it.
  7. The limitations of the study should be stated in discussion section: low number of included subject, large interindividual differences, 2D analysis of capillarisation
  8. Since the muscle fibre types and skeletal muscles could accumulate lipids differentially (DOI: 1242/jeb.217117 , DOI: 10.17305/bjbms.2021.5876 ), do you think that different more slow-twitch muscle (e.g. soleus) would exhibit the same behaviour as studied vastus lateralis muscle?
  9. Was two-way ANOVA used for statistics (one factor age and other pre/post exercise)? Where the changes were not concordant between groups, the interaction should be also reported, especially due to small number of included subjects.
  10. How were SDH and COX optical densities measured? Was any optical filter used for imaging? Was any colour deconvolution used? Did you measure optical density in a whole fibre or just part of the fibre since these staining are usually inhomogeneous throughout the fibres?
  11. Can you also provide representative images of SDH and COX stained sections? What was the time of incubation for these reactions.

Author Response

We thank the Supervising Editor and the Reviewer’s for their valuable comments regarding our investigation. We have revised the manuscript appropriately and feel is has strengthened this study. Specific responses can be found below which address each individual comment.

Reviewer 3

Comments and Suggestions for Authors

The authors studied the effects of single bout of prolonged moderate-intensity exercise (24h treadmill run) on proteomics, muscle fibre morphometry, intramyocellular lipid droplet content, mitochondrial oxidative activity, extracellular matrix and capillarisation. The paper is very interesting and has merit. I have a few comments:

1. The authors studied cross-sectional areas of muscles fibres. CSA is very sensitive to potential oblique sections, which can falsely increase CSA. Since the number of subjects is low, even a few obliquely cut sections could be important confounders. I suggest to also add the data on muscle fibre diameter, especially minimal Ferret diameter which is the least sensitive to the angle of section.

Answer 3.1 - The morphometry of muscle fibers was determined using the image processing software Visilog-6.9, which provides for each fiber its area, perimeter and shape factor. Unfortunately, no data on Ferret diameter is available. However, the shape factor, which is the reciprocal of circularity, was determined for each fiber. For this shape factor, a value of 1.0 indicates a circle, while a value >1.0 indicates an increasingly elongated ellipse, e.g. obliquely cut sections. This shape factor was measured for each fiber and as shown in Fig. S1B, it did not differ between fiber-types, between 37Y and 50Y groups pre- and post-24TR. There was thus no evidence that obliquely cut sections could be a significant confounder.

2. There is thought to be inverse correlation between muscle fibre size and oxidative capacity regardless of the muscle fibre type. It was also shown that large and small fibres can differentially response to pathologic and physiologic conditions (DOI: 1007/s00418-019-01810-7 ). Therefore, the analysis of different parameters (SDH, COX, capillarization, LAI) should be also performed separately for large and small fibres, regardless of the fibre type.

Answer 3.2 - The Reviewer suggests that we perform the analyses according to fiber size instead of fiber type. However, while specific antibodies precisely define fiber type, there is not clear threshold to identify a fiber as either small or large. Despite the absence of size threshold, we calculated Pearson’s correlations (r) between fiber CSA and several parameters, regardless of fiber type for 37Y and 50Y muscles PRE and POST 24TR:

COX

SDH

LAI

LC/PF

37Y

PRE

0.277

0.170

0.458

0.184

37Y

POST

0.336

0.195

-0.082

0.188

50Y

PRE

0.410

-0.028

0.108

0.321

50Y

POST

0.214

0.056

-0.047

0.303

As shown in the table above, we could essentially detect weak positive correlations. In contrast to the Reviewer’s suggestion, in our experimental model there was no evidence for an inverse correlation between muscle fibre size and oxidative capacity. Moreover, there was no evidence that CSA correlated with capillarization (LC/PF) or LAI.

3. The authors report that the capillarization increased after 24h of exercise. Since 24h is very short time, it is very unlikely that the capillary network would increase so substantially in this time. It is even less likely that the 2D method of studying capillarization, which has low sensitivity and specificity for detecting small changes (DOI: 1016/j.mvr.2009.11.005 ), would be able to show these changes in so few subjects. I believe the most possible explanation is that due to extreme 24h exercise, the muscle was depleted of energy sources and ATP, which could cause that rigor mortis in biopsy sample in a few minutes. In rigor mortis, the muscle fibres contract, and the capillary network around them becomes falsely enlarged.

Answer 3.3 – We thank the Reviewer for this suggestion. Indeed, for master athletes (50Y), a reduced oxidative capacity (Fig. S1D) might partly results in ATP depletion and fiber atrophy (Fig. 1C). This point is now indicated in the manuscript (lines 360-361):

“Whether alteration in energy metabolism partly account for fiber shrinkage remains to be established”

However, despite a reduced oxidative capacity and fiber atrophy, no enlargement of the capillary network occurred for master runners POST-24TR (Fig. 4B). In contrast for middle-aged athletes (37Y), there was enlargement of the capillary network, but no evidence for reduced oxidative capacity (Fig. S1D) or fiber atrophy (Fig. 1C) POST-24TR. In the manuscript, we rather suggest that intussusceptive angiogenesis might be involved (lines 312-313).

Indeed, angiogenesis can occur through (at least) two alternative mechanisms: sprouting (SA) and intussusceptive (or splitting) angiogenesis (IA). SA is a relatively slow process relying largely on cell proliferation. In contrast, IA is fast, does not primarily need cell proliferation, and can expand all existing capillary networks within hours (Burry et al. 2004; de Spiegelaere et al. 2012; Mentzer & Konerding 2014).

The organism switches from SA to IA during development, and IA prevails in adult life, being implicated in vascular remodeling. IA is a widespread phenomenon occurring in most tissues, including skeletal muscle (Egginton et al. 2001; Gianni-Barrera et al. 2013; Mentzer & Konerding 2014; Gianni-Barrera et al. 2018). Hemodynamic forces and shear stress, that occur during exercise, have a major influence in initiating this process (de Spiegelaere et al. 2012; Mentzer & Konerding 2014).

Blood vessel formation by intussusception has major advantages over sprouting: IA is faster than SA, and IA occurs without interfering with the local physiological conditions because no blind ending capillary segments are formed. In addition, endothelial migration and proliferation are kept to a minimum as the endothelial cells do not necessarily proliferate but rather increase in size and flatten. This results in a relatively lower metabolic cost of IA in comparison with SA.

Therefore, we agree with the Reviewer that sprouting angiogenesis that relies on cell division is unlikely to account for lengthening of capillaries. We also recognize that additional studies are required to identify the processes involved. However, we speculate that mechanisms such as intussusceptive angiogenesis (IA) might be involved in capillary remodeling and in part account for the 20-25% increase in surface of exchange between capillaries and muscle fibers that we observed only for 37Y runners after 24TR. This point is mentioned in the Discussion section (lines 312-313):  

“Whether mechanisms such as intussusceptive angiogenesis (Mentzer & Konerding 2014) are involved in capillary remodeling remains to be specified.” 

Burry et al. (2004) Developmental Dynamics 231, 474

De Spiegelaere et al. (2012) J Vasc Res 49, 390

Egginton et al. (2001) Cardiovascular Research 49, 634

Gianni-Barrera et al. (2013) Angiogenesis 16, 123

Gianni-Barrera et al. (2018) Angiogenesis 21, 883

Mentzer & Konerding (2014) Angiogenesis 17, 499

4. The authors refer to LC/PF index in abstract and discussion as tortuosity. Since this index is very limited index of tortuosity (note that true tortuosity and branching could be really evaluated only in longitudinal sections or 3D analysis), they should rather refer to it as it is LC/PF index, so that readers that are not expert in 2D capillary analysis or stereology are not misled.

Answer 3.4 - We agree with the Reviewer and instead of tortuosity we now refer to “capillary-to-fiber interface” in the abstract, and to “LC/PF index” in the Discussion.

5. For the same reasons as above, I find it unlikely that the endomysium would actually increase in diameter in 24h due to synthesis of collagen (which is stained by Sirius red). Could the noted increase be the consequence of maybe oedema?

Answer 3.5a - The biopsies used in the present study were acquired in a protocol previously described (Martin et al. 2010; Millet et al. 2011). In these reports, we showed that a marked dehydration, able to cause marked ionic imbalance and perturbations in muscle membrane excitability, was unlikely to occur. Indeed, exercise intensity was relatively low and no significant change in plasma [K+] and [Na+] was observed during 24TR. In line, the M-wave amplitude measured for the vastus lateralis muscle, reflecting membrane excitability, did not significantly change after 24TR neither for 37Y (49.9 + 5.3 mV for PRE and 50.5 + 6.2 mV for POST, P=0.793), nor for 50Y (46.1 + 8.3 mV for PRE and 37.6 + 8.7 mV for POST, P=0.197).

This point is now better specified in the Discussion section (lines 291-294):

“Whether ECM thickening is related to marked dehydration is unlikely, as our previous data indicated that 24TR is a relatively low-intensity exercise that does not involve ionic imbalance or perturbation in membrane excitability [24, 28].”

 Also is there any significant statistical interaction between age and pre and post exercise endomysium thickness (two-way ANOVA)?

Answer 3.5b – Our data correspond to an experiment in which two groups (37Y and 50Y) of runners have been followed at two different times of a treatment (PRE- and POST-24TR). Therefore and as indicated in Materials and Methods (§4.9), “Differences between groups were assessed using repeated measures ANOVA, followed by post-hoc Fisher’s test for pairwise comparisons between groups”. We also performed two-way ANOVA and found a statistical significant (P = 0.031) interaction between time and group.

6. Authors use lipid accumulation index (LAI) to measure lipid content in the muscle fibres. Since it does not measure accumulation of lipids, but more of lipid content, the lipid content index would be more appropriate name for it.

Answer 3.6 - We agree with the Reviewer and we now refer to “lipid content index” instead of lipid accumulation index (LAI).

7. The limitations of the study should be stated in discussion section: low number of included subject, large interindividual differences, 2D analysis of capillarisation

Answer 3.7 - A limitation section is now included at the end of the discussion (lines 375-380):

“The main limitation of our study is that we could only include a low number of athletes. Further works are also required to investigate the influence of gender. More experimental studies are also necessary to perform 3D analyses of capillarization and to characterize the intramyocellular localization and composition of the lipid droplets. We believe that the present paper is an important contribution, but it must be seen as a first step toward identification of early signs of muscle aging.”

8. Since the muscle fibre types and skeletal muscles could accumulate lipids differentially (DOI: 1242/jeb.217117 , DOI: 10.17305/bjbms.2021.5876 ), do you think that different more slow-twitch muscle (e.g. soleus) would exhibit the same behaviour as studied vastus lateralis muscle?

Answer 3.8 - The vastus lateralis and soleus muscles have different daily loading demand and display divergent cellular contractile properties [Luden et al. (2008) Am J Physiol Regul Integr Comp Physiol 295: R1593]. Due to chronic loading the soleus is better conditioned to exercise than the vastus lateralis, and for example, does not increase protein synthesis to the magnitude of the vastus lateralis following exercise [Trappe et al. (2004) Acta Physiol Scand182: 189]. Thus, we may not expect exactly the same behavior, although it is difficult to make an assessment without any data on that muscle.

9. Was two-way ANOVA used for statistics (one factor age and other pre/post exercise)? Where the changes were not concordant between groups, the interaction should be also reported, especially due to small number of included subjects.

Answer 3.9 – Because our data correspond to an experiment in which 2 groups (37Y and 50Y) of runners have been followed at 2 different times of a treatment (PRE- and POST-24TR), differences between groups were assessed using repeated measures ANOVA, followed by post-hoc Fisher’s test for pairwise comparisons between groups. This is indicated in Materials and Methods section (§4.9).

There was a statistically significant interaction between time and group for type IIA CSA (p = 0.049; Fig. 1C), type-I LAI (p = 0.037; Fig. 2B), number of lipid droplets per type-I fiber (p = 0.045, Fig. 2C), ECM thickness (p = 0.031; Fig. 3B), type-I LC/PF (p = 0.001; Fig. 4B left), and type-IIA LC/PF (p = 0.003; Fig. 4B right). This is now indicated in each Figure legend.

10. How were SDH and COX optical densities measured? Was any optical filter used for imaging? Was any colour deconvolution used? Did you measure optical density in a whole fibre or just part of the fibre since these staining are usually inhomogeneous throughout the fibres?

Answer 3.10 – Mitochondrial histoenzymology inadvertently refers to a wrong reference. The correct reference [Merlet et al. (2020) Am J Hematol 95, 1257] is now indicated (line 426). We are sorry for that and we double-checked all references in the manuscript. No optical filter and no colour deconvolution was used for SDH and COX optical densities measurements. SDH and COX optical density was measured for the whole fiber, and images were matched with MHC (fiber type) to specify the type of each fiber.

11. Can you also provide representative images of SDH and COX stained sections? What was the time of incubation for these reactions.

Answer 3.11 – Representative images of SDH and COX stained sections are now provided in Figure S1. We used 2h incubation at 37°C for SDH and COX activities.

Round 2

Reviewer 3 Report

The authors addressed most of my comments satisfactorily. I have just a few additional minor comments.

  1. Now that the authors changed lipid accumulation index to lipid content index, the abbreviation LAI should also be changed or omitted.
  2. Were the data distributed normally? What test was use to test it? Where there any outliers (more than 2 SD away from the mean)? How do results change if the outliers are removed?

Author Response

The authors addressed most of my comments satisfactorily. I have just a few additional minor comments.

  1. Now that the authors changed lipid accumulation index to lipid content index, the abbreviation LAI should also be changed or omitted.

Answer-1. In agreement with the Reviewer, we changed all LAI to LI.

  1. Were the data distributed normally? What test was use to test it? Where there any outliers (more than 2 SD away from the mean)? How do results change if the outliers are removed?

Answer 2. The Shapiro–Wilk test has been used to verify normality. All Shapiro-Wilk tests were greater than the significance level alpha=0.05 (CSA-IIA: 0.272; LI-I: 0.372; nb LD/fiber-I: 0.375; ECM thickness: 0.712; LC/PF-I: 0.793; LC/PF-IIA: 0.749), and therefore all data that we tested were distributed normally.

No outlier was detected for any variable for 37Y or 50Y, PRE or POST-24TR.